



# Top Level Rotor Optimisations based on Actuator Disc Theory

Peter Jamieson[1]

[1]Centre for Doctoral Training in Wind and Marine Energy, University of Strathclyde, Glasgow, G1 1XW, UK

*Correspondence to*: P Jamieson (peter.jamieson@strath.ac.uk)

**Abstract**. Ahead of the elaborate rotor optimisation modelling that would support detailed design, it is shown that significant insight and new design directions can be indicated with simple, high level analyses based on actuator disc theory. The basic equations derived from actuator disc theory for rotor power, axial thrust and out of plane bending moment in any given wind condition involve essentially only the rotor radius, $R$, and the axial induction factor, $a$. Radius, bending moment or thrust may be constrained or fixed with quite different rotor optimisations resulting in each case. The case of fixed radius or rotor

diameter leads to conventional rotor design and the long-established result that power is maximised with an axial induction factor, $a = 1/3$. When the out of plane bending moment is constrained to a fixed value with axial induction variable in value (but constant radially) and when rotor radius is also variable, an optimum axial induction of $1/5$ is determined. This leads to a rotor that is expanded in diameter 11.6% gaining 7.6% in power and with thrust reduced by 10%. This is the "low induction rotor" which has been investigated by Chaviaropoulos (2013). However, with an optimum radially varying

distribution of axial induction, the same 7.6% power gain can be obtained with only 6.7% expansion in rotor diameter. When without constraint on bending moment, the thrust is constrained to a fixed value, the power is maximised as $a \rightarrow 0$ which for finite power extraction would require $R \rightarrow \infty$. This result is relevant when secondary rotors are used for power extraction from a primary rotor. To avoid too much loss of the source power available from the primary rotor, the secondary rotors must operate at very low induction factors whilst avoiding too high a tip speed or an excessive rotor diameter. Some general

design issues of secondary rotors are explored. It is suggested that they may have most practical potential for large vertical axis turbines avoiding the severe penalties on drive train cost and weight implicit in the usual method of power extraction from a central shaft.

## 1 Introduction

Basic actuator disc equations for power, thrust and out-of-plane bending moment as related to ambient wind speed, $U_0$, air

density, $\rho$, and rotor radius, $R$, are presented in Table 1. The coefficients of power and thrust, $C_p$, and $C_t$, depend only on axial induction factor, $a$, and are in widespread use. A companion out-of-plane bending moment coefficient, $C_m$ is also defined as in Jamieson (2011). The standard assumption of blade element momentum theories is that each annular ring of the actuator disc can be treated as independent. Thus, when the axial induction varies radially, rotor area averaged values of the coefficients may be defined as in the right-hand column of Table 1.





**Table 1.** Basic actuator disc equations for power, thrust and out of plane bending moment

| Variable | | Actuator Disc Equation | Rotor power coefficient (radially constant axial induction) | Rotor power coefficient (radially variable axial induction) |
|---|---|---|---|---|
| Power | $P$ | $P = 0.5\rho U_0^3 \pi R^2 C_p$ | $C_p = 4a(1-a)^2$ | $C_p = 8\int_0^1 a(1-a)^2 x\,dx$ |
| Thrust | $T$ | $T = 0.5\rho U_0^2 \pi R^2 C_t$ | $C_t = 4a(1-a)$ | $C_t = 8\int_0^1 a(1-a)x\,dx$ |
| Moment | $M$ | $M = 0.5\rho U_0^2 \pi R^3 C_m$ | $C_m = \dfrac{8}{3}a(1-a)$ | $C_m = 8\int_0^1 a(1-a)x^2\,dx$ |

Three distinct optimisations are now considered with the objective in each case of maximising power;

5    a)   The rotor radius is fixed and axial induction is to be determined

    b)   The out of plane bending moment is fixed but rotor radius and axial induction are variable

    c)   The rotor thrust is fixed but rotor radius and axial induction are variable

**2 Optimisations with radially constant induction**

The optimisations are first considered in the context of an axial induction that does not vary spanwise. Case a) is the familiar

10   one where, with radius $R$ fixed, power, $P \propto a(1-a)^2$ and is consequently maximised with $a = 1/3$. This represents conventional design and is the basis of a reference design used in subsequent comparisons. In the reference design, $R = R_0$, $P = P_0$, $T = T_0$ and $M = M_0$ where the reference values, $P_0, T_0, M_0$, are all based on $R = R_0$ and $(a = a_0 = 1/3)$. In Case b), the out-of-plane blade bending moment is fixed and: $M = M_0 = 0.5\rho U_0^2 \pi R^3 C_m = \frac{4}{3}\rho U_0^2 \pi R^3 a(1-a) = \frac{8}{27}\rho U_0^2 \pi R_0^3$ which on solving for $R$ yields;

$$R = \left\{ \frac{3M_0}{4\rho U_0^2 \pi a(1-a)} \right\}^{1/3} \tag{1}$$

15   Substituting for $R$ from Eq. (1), the power equation, $P = 0.5\rho U_0^3 \pi R^2 C_p = 2\rho U_0^3 \pi R^2 a(1-a)^2$ becomes;

$$P = 2\rho U_0^3 \pi \left\{ \frac{3M_0}{4\rho U_0^2 \pi a(1-a)} \right\}^{2/3} a(1-a)^2 \tag{2}$$

From Eq. (2), the power, $P$ now varies only with $a$ and;

$$P \propto a^{1/3}(1-a)^{4/3} \tag{3}$$





Differentiating $P$ in Eq. (3) to find a maximum leads to $(1-a)(1-5a) = 0$ and hence $P$ is maximised at $a = \frac{1}{5}$.

Comparing with a standard rotor design, when $a = \frac{1}{5}$ and $P$ is maximum:

$$\frac{R}{R_0} = \left\{ \frac{a_0(1-a_0)}{a(1-a)} \right\}^{1/3} = 1.116 \tag{4}$$

$$\frac{P}{P_0} = \frac{a(1-a)^2}{a_0(1-a_0)^2} \left( \frac{R}{R_s} \right)^2 = 1.076 \tag{5}$$

$$\frac{T}{T_0} = \frac{a(1-a)}{a_0(1-a_0)} \left( \frac{R}{R_s} \right)^2 = 0.896 \tag{6}$$

As in Jamieson (2018), general trends of $R$, $P$, $M$ and $T$ relative to unit values of the standard rotor are presented in Fig. 1.

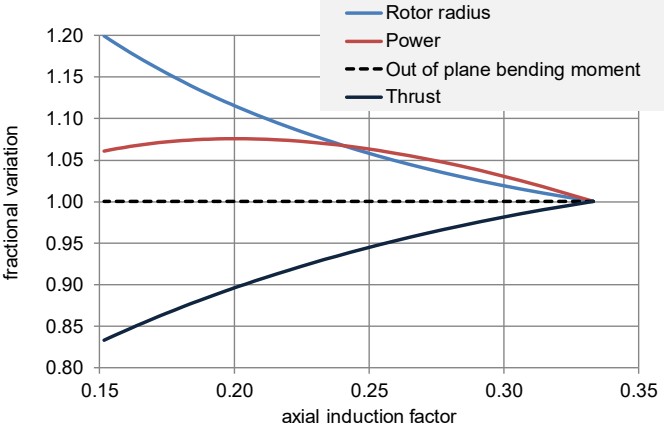

**Figure 1: Design parameters related to axial induction.**

The analysis indicates that a rotor designed for an axial induction factor of 0.2 that is 11.6% larger in diameter can operate with 7.6% increased power and 10% less thrust yet at the same level of blade rotor out-of-plane bending moment as the baseline design. In case (c), the thrust is maintained at a constant value, $T_0$. Since power $\propto R^2 a(1-a)^2$ and $T \propto R^2 a(1-a)$ is constant, it is evident that the power, $P \propto (1-a)$, and is maximised as $a \to 0$. However, for the power to be finite and positive when the axial induction and hence the power coefficient are zero requires $R \to \infty$.



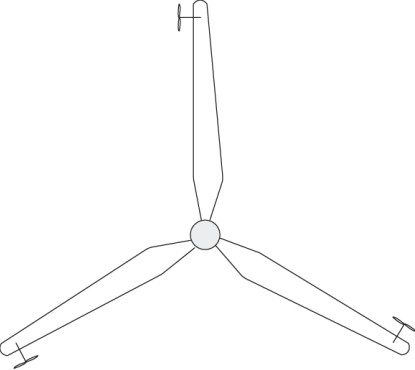

**Figure 2: Rotor with secondary rotors.**

As opposed to the conventional solution of power take-off from a central shaft, additional (secondary) rotors are set on the blades or other support arms at a radial distance from the central axis of the primary rotor thereby experiencing a high

relative wind speed. The ideal optimisation at zero induction and hence infinite radius cannot be realised but it will be shown that very low induction values are feasible without unacceptably large secondary rotors. The secondary rotor may be therefore be considered as an "ultra-low induction rotor". In the system of Fig. 2, the torque reaction to the primary rotor is provided by thrust on the secondary rotors and a specific value of thrust on each secondary rotor is therefore required to optimise power extraction from the primary rotor. The secondary rotors are small, high speed rotors and the sum of design

torques of all secondary rotors can be much less than the design torque associated with power take-off in the conventional way from a central shaft. This property can offer a solution to a key problem of large VAWT design where an inherently lower shaft speed than any equivalent HAWT puts a large premium on drive train torque, mass and cost.

### 3 Low induction rotor design

For a radially constant axial induction distribution and fixed out-of-plane bending moment, $M = M_0$, it was established in

Section 2 that , $a = 0.2$ maximises power giving a 7.6% power gain for 11.6% radius expansion compared to conventional design. The question then arises if an optimised radially varying distribution of axial induction can realise greater power gains or, for example, the same 7.6% power gain at reduced rotor expansion. Related to this is the question of what may be a suitable, efficient generalised model of the radially variable axial induction. A representation in the form $a(x) = a(1 - x^n)^p$ is found to be versatile and highly effective. With arbitrary values of only 2 free variables, $n$ and $p$, a wide range of

distributions can be generated (Fig. 3). This even includes approximations to constant values of axial induction less than 0.333 for example, $a = 0.2$. The curve (yellow trace) of Fig. 3 illustrates this although a much more accurate approximation

than shown can be obtained. More general optimisation methods could be employed to determine optimum distributions of axial induction subject to varied constraints but the simple approach adopted here is highly effective.

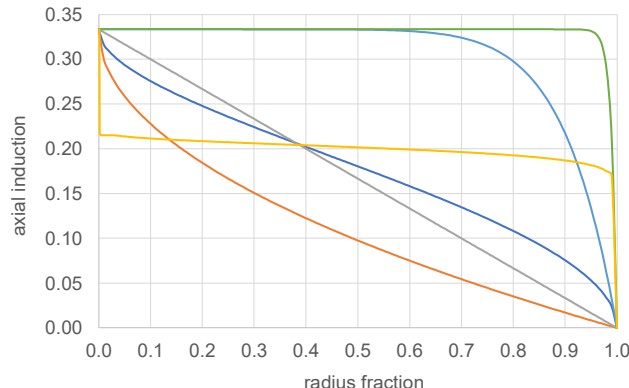

**Figure 3: Distributions of axial induction for arbitrary choices on $n$ and $p$**

Now there can never be benefit in $a > 1/3$ as the bending moment would be increased and power decreased. Also as $x \to 0$, the bending moment, $M \to 0$ and so in the limit $x \to 0$, that is approaching the shaft centre, it is logical that $a \to 1/3$ in any design that seeks to constrain only bending moment. In the following analyses, $a, n$ and $p$ are all treated as free variables although, as expected, the value determined for $a$ is usually very close to $1/3$. This tends to confirm that the optimisation, although in effect having only two free variables, $n$ and $p$, is quite accurate. Polynomial representations by comparison are

far inferior. A quadratic, for example, $a_2 x^2 + a_1 x + a_0$, with $a_0 = 1/3$, would have 2 free variables, $a_2$ and $a_1$, but could only represent linear or parabolic shapes. In order to have results that are likely to be realistic for typical rotors with small finite blade numbers, a tip loss effect is introduced using the Prandtl tip loss factor, $F(x) = \frac{2}{\pi} acos\left\{ e^{-\frac{(1-x)B\lambda}{2(1-a)}} \right\}$. The question of an overall maximum in power regardless of required diameter expansion is now addressed. Using the generalised forms of $C_p$ and $C_m$ from Table 1, the power is expressed as;

$$P(a, n, p) = \frac{4\rho\pi U_0^3 M_0^{\frac{2}{3}} \int_0^1 a(1 - x^n)^p \{1 - a(1 - x^n)^p\}^2 \, x \, F(x) dx}{\left\{ 4\rho\pi U_0^2 \int_0^1 a(1 - x^n)^p \{1 - a(1 - x^n)^p\} x^2 \, F(x) dx \right\}^{2/3}} \tag{7}$$

Using a maximisation routine such as available in PTC Mathcad 15, an overall maximum in power $P(a, n, p)$ is obtained with values; $a = 0.331, n = 1.504, p = 1.125$ giving an axial induction distribution as in Fig 4. The gain in power (see Fig. 4, $P_{max}$) is found to be 11.9% much greater than the 7.6 % for a radially constant axial induction but requiring a radial expansion of 34%. This is too large a radial expansion to be of practical benefit considering the implications in increased tip speed or drive train torque. In the next analysis the radial expansion is constrained (see Fig. 4, $P_{con}$) to a value such that the

power gain is 7.6% as for optimum constant induction. The associated axial induction distribution has parameters; $a =$



$0.333, n = 0.417, p = 0.136$ as illustrated in Fig. 4. Note that all the distributions of Fig. 4 maintain the same constant value of out-of-plane bending moment at shaft centreline. The striking result however is that this same power gain of 7.6% is realised with a radius expansion of only 6.7% (diamond marked point of Fig.5) as opposed to the 11.6 % (triangular marked point of Fig. 5) required with a constant axial induction of 0.2.

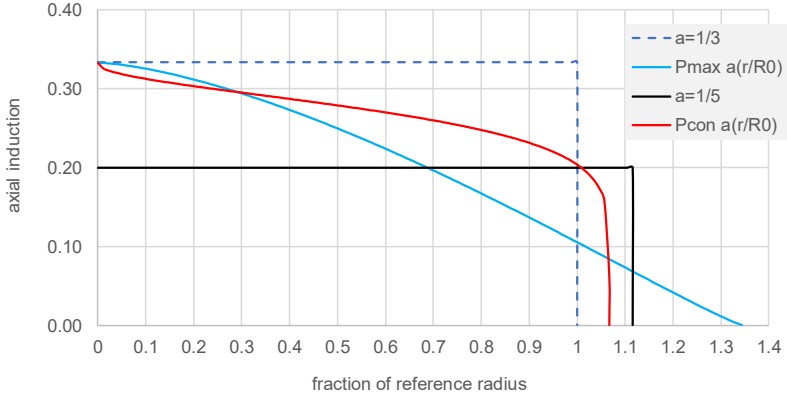

**Figure 4: Axial induction distributions giving rise to the same out-of-plane bending moment, $M_0$ , at rotor centre.**

Also shown in Fig. 5 is the ratio of power gain to expansion ratio which maximises around 3% expansion. Above this low level, the required rotor expansion rises more rapidly than the gain in power although, the most economic benefit will probably arise with power gains and rotor expansions in a 5% to 10% range.

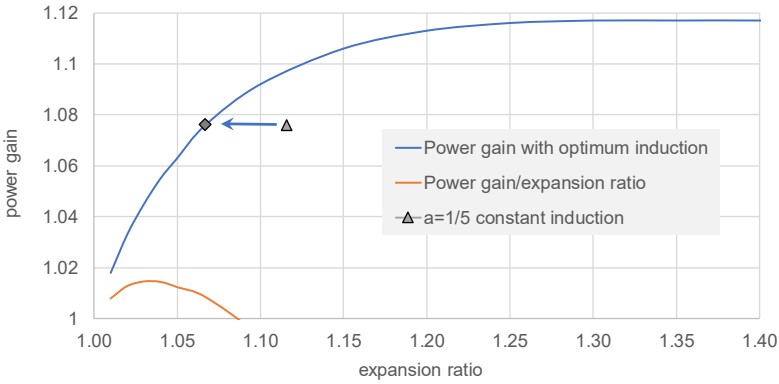

**Figure 5: Power gain related to rotor radius expansion ratio.**

Comparing (see Fig. 4) the optimum axial induction distribution (for 7.6% power gain) with the constant value of 0.2, it is evident that more power is being obtained over most of span except near the blade tip. Consistent with these higher power levels, there is only a 3.5% reduction in axial thrust for the radially variable axial distribution ($P_{con}$ in Fig. 4) as opposed to





approximately 10% reduction for constant induction at $a = 0.2$. If cost of energy modelling suggests that there is benefit say from reduced wake impacts in a thrust reduction greater than 3.5%, say at the same power gain of 7.6%, with appropriate constraints on the power maximisation procedure, the necessary rotor expansion can then be related to thrust reduction as in Fig. 6.

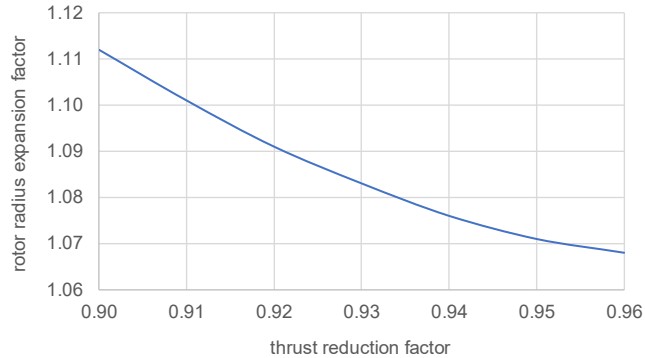

**Figure 6: Rotor expansion related to thrust reduction for a fixed power gain (7.6%)**

Tip loss has no effect in comparing distributions where the axial induction is constant radially because it cancels in the power, moment and thrust ratios providing the low induction rotor is compared with a reference rotor having the same number of blades. It has a small effect (Fig. 7) for designs with rotor expansions below about 15% and a more noticeable
10 effect at large expansion ratios which however may be of little practical interest.

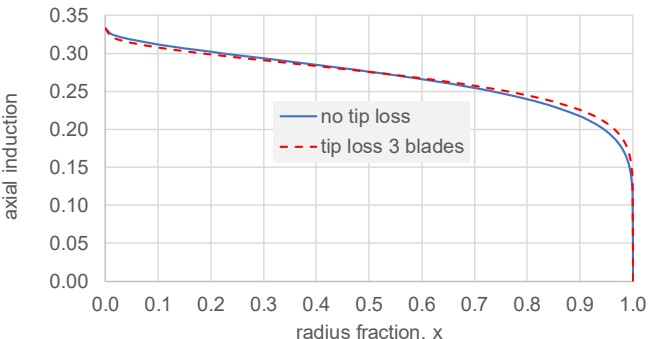

**Figure 7: Effect of tip loss on optimum axial induction distributions for a power gain of 7.6%.**

The distributions in Fig. 7 are very similar and that is what matters most. On account of the sensitivity of the power law relationships, the associated values of $n$ and $p$ will often differ considerably. For no tip loss $n = 0.416$ and $p = 0.136$; with
15 tip loss $n = 0.295$ and $p = 0.112$. Another main issue of practical relevance is that blades are never aerodynamically active near the shaft centreline. They may become cylindrical near the root contributing only drag and connect to a hub having a



conical cover or spinner. To approximate the loss of aerodynamic performance in the hub area, some of the analyses were repeated with lower limits on integrals such as in Eq. (7) changed from 0 to 0.15. As with tip loss, effects were only very noticeable at large (impractical) expansion ratios. Fig. 8 compares the results for maximum possible power gain with and without exclusion of the first 15% of span.

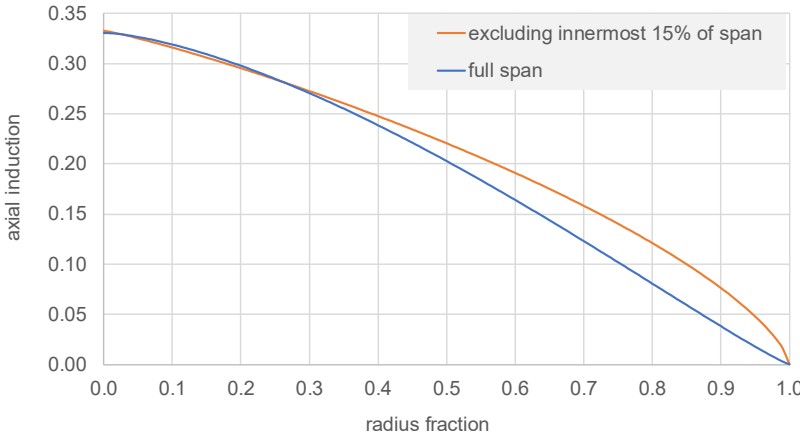

**Figure 8: Axial induction distributions for maximum power gain**

Table 2 presents data relating to axial induction distributions of Fig. 8. Although the power gains differ only ~ 1%, there is a noticeable difference in the axial induction distributions of Fig. 7 and a large difference in the rotor expansions at 34% for the complete span being aerodynamically active and 25% when the innermost 15% of span is excluded. When designs in a

10   more realistic range of parameters are considered, for example, as in Fig 7 with power gain restricted to 7.6%, there is no significant difference between cases with and without exclusion of the inner 15% of the rotor.

**Table 2.** Parameters of the distributions for maximum power gain

| Fraction of span inactive aerodynamically | 0 | 0.15 |
|---|---|---|
| a | 0.331 | 0.333 |
| n | 1.504 | 1.130 |
| p | 1.125 | 0.674 |
| Radius expansion factor | 1.343 | 1.246 |
| Power gain | 1.119 | 1.109 |

15   Simple actuator disc theory determines that; $\frac{dC_t}{da} = \frac{4}{3}$ when $\frac{dC_p}{da} = 0$. There is therefore a relatively large reduction in thrust and associated bending moments to be gained from reducing induction levels a little below the theoretical optimum for maximum power of $a = 1/3$ and independent blade manufactures have long been aware of this. More radical optimisations



of low induction designs with expanded rotor diameter have only recently been explored by Chaviaropoulos (2013), Bottasso (2014), Madsen (2015) and Quinn (2016). Overall the present results show that, even using simple actuator disc theory, there may be great value in treating the axial induction distribution and rotor diameter as free variables in a basic system optimisation for lowest cost of energy where direct power gains, rotor loading, reduced wake effects from thrust reduction

can all be traded in the design optimisation.

## 4 Secondary rotor design

### 4.1 Introduction

The secondary rotor concept has been considered previously by Watson (1988), St-Germain (1992), Jack (1992) and Madsen (2008). Jamieson (2011) mentioned it as a possible solution to the design challenge faced by large VAWTs where a very low

optimum speed leads to high drive train torque, weight and cost if the power is extracted in the most usual way from the central shaft. Leithead (2019) employs secondary rotors for power take-off in an innovative X-rotor VAWT design. The design tip speed of a large offshore HAWT may be 2 or 3 times greater than the 40 - 45 m/s typical of a VAWT. Secondary rotors near the tip of HAWT blades will therefore experience a much higher relative flow velocity and may thus be smaller in diameter than those of a VAWT of similar rated power. However the tip region of a large HAWT is subject to large

deflections and a torsional stiffness that is relatively reducing with upscaling. Thus reacting the total edgewise load of a blade near the tip may pose problems for aerodynamic stability and structural stiffness. Even more problematic may be preserving alignment of secondary rotors on a pitching blade. The classic issues with VAWTs which had led to them being uncompetitive historically are a) an intrinsically lower optimum speed leading to factors of 2 or 3 on drive train torque, weight and cost and b) reduced power performance associated with intrinsically lower average lift to drag ratios per cycle of

rotation leading to maximum power coefficients ~ 0.4 when large HAWTs have power coefficients ~0.5. Power take-off using secondary rotors may provide a much more effective drive train solution and breath new life into VAWT technology.

### 4.2 Power extraction using secondary rotors

The focus in the following analyses is on secondary rotors for a primary rotor of VAWT design although much of the analyses are directly relevant or easily adapted to HAWT design. The secondary rotors are always assumed to be HAWTs. In

the following analyses upper case symbols refer to a primary rotor and lower case to a secondary rotor. Where there are multiple secondary rotors, the parameters of 1 of $n$ rotors will have the subscript $n$. The aerodynamic torque of the primary rotor is reacted by the total thrust of the secondary rotors acting (under the present simplified assumptions) with a moment arm at the maximum radius $R_0$ of the primary rotor. The relative wind speed incident on the secondary rotors is equal to the tip speed of the VAWT ~ 40 m/s and as a further simplification, the ambient wind speed which is small in comparison is

ignored. The power generated by the primary rotor is then;





$$P = nt_n \Omega R_0 \qquad (8)$$

The total power, $p$, extracted by the secondary rotors is then;

$$p = nt_n \Omega R_0 (1 - a) \qquad (9)$$

Now with the usual assumption that each annular ring of the actuator disc can be analysed independently, then Eq. (9) applies to the elemental power and thrust contributions of each annulus and a radially varying axial induction, $a(x)$, will have exactly the same performance as a constant induction of $\bar{a}$, the area averaged value of $a(x)$. For this reason only

5    radially constant values of axial induction are considered although, in a detailed design embracing all aspects of structure and loads, there may be some benefits from radially varying axial induction. This result is of course quite different from the previous case where the bending moment is constrained and radial variation of axial induction is very significant. As an example, to focus discussion of secondary rotor design issues, parameters as in Table 3 are selected for a VAWT rated at 5 MW.

**Table 3.** Parameters of a primary H type VAWT rotor and of secondary HAWT rotors

| | | Primary | | Secondary | Unit |
|---|---|---|---|---|---|
| Ambient wind speed | $U_0$ | | | | m/s |
| Rated power | $P$ | $5/\{\eta(1-a)\}$ | $p_n$ | $5/n$ | MW |
| Design tip speed | $V_t$ | 40 | $v_t$ | 160 | m/s |
| Rotor power coefficient | $C_P$ | 0.4 | $C_p$ | $4a(1-a)^2$ | |
| Rotor thrust coefficient | $C_T$ | | $C_t$ | $4a(1-a)$ | |
| Rotor radius | $R_0$ | 65 | $r_n$ | | m |
| Rotor angular speed | $\Omega$ | | $\omega_n$ | | rad/s |
| Design tip speed ratio | $\Lambda$ | 4 | $\lambda$ | $\omega r_n / \Omega R_0 = 4$ | |
| Blade length | $L$ | 100 | | | m |
| Drive train efficiency | | | $\eta$ | | |
| Number of rotors | | 1 | $n$ | 6 | |
| Rotor thrust | $T$ | | $t_n$ | | N |
| Rotor torque | $Q$ | $nR_0 t_n$ | $q_n$ | $p_n/\omega_n$ | Nm |
| Blade chord | | | $c_n$ | | m |

**4.3 Sizing of secondary rotors**

Now the ratio of radius of one of $n$ secondary rotors to that of the primary rotor can be expressed as;

$$\frac{r_n}{R_0} = \left\{ \frac{LC_P}{2n\pi\lambda^3 a(1-a)R_0} \right\}^{0.5} \qquad (10)$$





The ratio of secondary to primary rotor radius defined by Eq. (10) is shown in Fig. 9 as based on the data of Table 3. The curve is symmetrical about $a = 0.5$ although this is not obvious as a logarithmic scale is employed in order to show more clearly the variation of $r_n/R_0$ at very low axial induction values. The vertical line of Fig. 9 marks $a = 0.333$. There is no interest in greater values of $a$ and the optimum design value for an effective system will certainly be much less than 0.333 as

this would imply a sacrifice of $1/3$ of primary rotor power. In the data of Table 3 the number of secondary rotors is chosen as 6 which may be 2 on each of 3 blades or 3 on each of 2. A value of $a$ of 0.05 is chosen for further illustration of secondary rotor design issues. This implies a sacrifice of 5% of primary rotor power and the associated radius fraction is 0.084. Each secondary rotor then has a radius ($\sim 5$ m) that is 8.4% of the primary rotor radius (60m).

### 4.4 Torque benefit of secondary rotors

A major issue with large VAWTs especially is a very high level of drive train torque. In a conventional drive train solution with power take-off from a central shaft, the torque, Q, of the primary rotor would drive mass and cost of the drive train. To assess the benefit in secondary rotor power take-off the ratio of the sum of secondary rotor torques to Q is now compared.

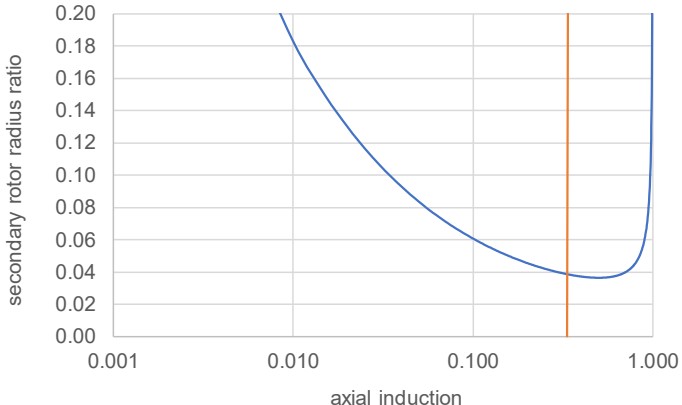

**Figure 9: ratio of secondary rotor radius to that of primary rotor**

$$\frac{nq_n}{Q} = \frac{p_n}{\omega_n Q} = \frac{(1-a)\Omega}{n\omega_n} = \frac{(1-a)r_n}{\lambda R_0} \tag{11}$$

For a design with $a = 0.05$ and parameters otherwise as in Table 3, the torque ratio $\frac{nq_n}{Q} = \frac{(1-a)r_n}{\lambda R_0}$ has a value $\frac{0.95 \times 0.084}{4} = 0.02$ showing that the sum of secondary rotor torques is $\sim 1/50$th of primary rotor torque. As a power take-off system, each secondary rotor system comprises both bearings and generator but also an aerodynamic rotor system. The

estimates of secondary rotor diameter and torque reduction factor are realistic providing it is accepted that at $a = 0.05$, the





fraction of available primary rotor power extracted will be less than 95% to an extent depending on the effect of parasitic drag losses. For conventional large HAWTs and possibly more so for VAWTs, rotor cost is generally less than the drive train cost but even at ratios $2/50^{th}$, $3/50^{th}$ or much more, there are potentially very large savings in cost and weight of power conversion with secondary rotors. The further benefits of multiple rotors are in $r_n$ reducing as $1/\sqrt{n}$ with the torque ratio of

Eq. (11) similarly reducing.

### 4.5 Design characteristics of secondary rotors

Does the design of the secondary rotor may differ much from conventional HAWT designs considering the unusually high relative wind speed and unusually low design levels of axial induction? This is initially assessed by deriving an equation for rotor solidity. From Jamieson (2011) a non-dimensional lift distribution, with $Cl_d$ as design lift coefficient (lift value at

maximum lift to drag ratio), is determined as;

$$\frac{c_n Cl_d}{r_n} = \frac{8\pi a(1-a)F(x)}{B\lambda(1+\acute{a})\sqrt{(1-a)^2 + \lambda^2 x^2(1+\acute{a})^2}} \tag{12}$$

In Eq. (12) the tangential induction factor, $\acute{a}$ , is determined as;.

$$\acute{a} \equiv \acute{a}(x) = \frac{(4a - 4a^2 + \lambda^2 x^2)^{0.5} - \lambda x}{2\lambda x} \tag{13}$$

Considering an annular ring of the rotor swept area of spanwise width, $dr$, the local solidity is the sum of planform elemental areas of $B$ blades within the ring as a ratio of the complete swept area of the ring. Thus the local solidity at radius $r$ is given as;

$$\sigma_n(r) = \frac{Bc_n dr}{2\pi r dr} = \frac{Bc_n}{2\pi r} \tag{14}$$

and the solidity of the whole rotor is then;

$$\sigma_n = \frac{2}{\pi r_n^2}\int_0^{r_n}\pi r\,\sigma_n(r)dr = 2\int_0^1 x\,\sigma_n(x)dx = 2\int_0^1 \frac{4a(1-a)F(x)}{x\lambda Cl_d(1+\acute{a})\sqrt{(1-a)^2 + \lambda^2 x^2(1+\acute{a})^2}}dx \tag{15}$$

The right hand side of Eq. (15) is obtained using Eq. 12 to substitute for $c_n$ in Eq. (14). Applying a tip loss factor, $F(x)$, appropriate to a 3 bladed rotor, omitting the inner rotor region where solidity would become infinite, taking a typical aerofoil with design lift coefficient, $Cl_d$, of 0.8 an estimate of secondary rotor solidity with a = 0.05 and otherwise consistent with the values of Table 3 is determined as;

$$\sigma_n = \int_{0.15}^1 \frac{8a(1-a)F(x)}{x\lambda Cl_d(1+\acute{a})\sqrt{(1-a)^2 + \lambda^2 x^2(1+\acute{a})^2}}dx = 0.072 \tag{16}$$

The dependence of rotor solidity on aerofoil design lift coefficient is illustrated in Fig. 10. An aerofoil such as NACA 63-418 has been used on wind turbines and (with some variation according to data sources) may provide a lift to drag ratio of ~125 at $Cl_d \sim 1$. According to Fig. 10 this may yield a solidity ~ 6% at a design axial induction ~ 0.05 which is only a little higher than values of 4%-5% most common in large HAWT designs. Thus the secondary rotor need not differ much from




conventional designs of large HAWTs in respect of solidity. Light loading from a very low design axial induction value and very high relative flow velocities have mutually compensating impacts on rotor solidity whereas a secondary rotor design for the usual design values of axial induction, $a \sim 1/3$, would have solidity $\sim 30\%$.

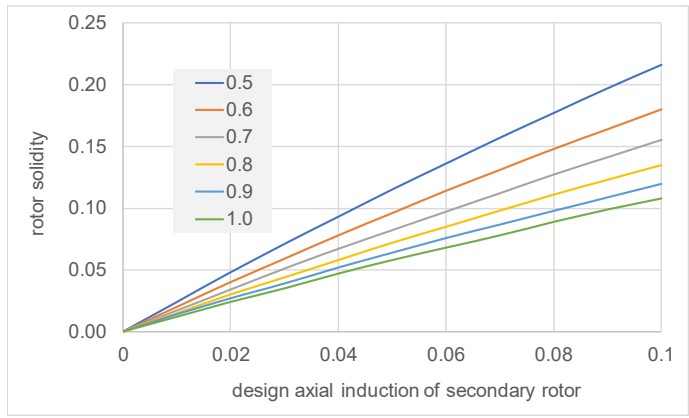

**Figure 10 Rotor solidity related to design axial induction and design lift coefficient**

The next consideration for secondary rotor design is the range of Reynolds number, $Re$. For a solidity $\sim 0.07$ as in Eq. (16), the chord at around 80% span will be;

$$c_n \sim \frac{\sigma_n r_n}{0.8B} = \frac{0.072 \times 5}{0.8 \times 3} = 0.15 \tag{17}$$

and the associated Reynolds Number is;

$$Re = \frac{0.8\rho v_t c_n}{\mu} = \frac{0.8 \times 1.225 \times 160 \times 0.15}{1.8 \times 10^{-5}} = 1306667 \tag{18}$$

Considering the high tip speed of the secondary rotor, using $v_t$ as the resultant velocity in the estimate of Eq. (18), and by implication neglecting the ambient wind speed, will give a good approximation. Equation (18) shows that $Re$ values of the secondary rotor will be in a normal range for medium to large HAWTs although the rotor diameter is small $\sim 10$ m.

Another important design consideration is the level of operational loads on the secondary rotor. Assuming a rated wind speed of $U_r = 11$ m/s, and a relative wind speed for the secondary rotors of 160 m/s then, compared to a conventional rotor of similar diameter, rotor thrusts and out-of-plane bending moments are both in the same ratio;

$$\frac{t_n}{T_0} = \frac{m_n}{M_0} = \frac{v_t^2 a(1-a)}{U_r^2 a_0(1-a_0)} = \frac{160^2 \times 0.05 \times 0.95}{11^2 \times 0.333 \times 0.667} = 45.2 \tag{19}$$

This is a huge increase in steady operational loading compared to conventional design. Also the steady and turbulent components of the ambient wind speed will introduce cyclic and random disturbances to secondary rotor inflow which may



increase available power (Leithead 2019) but will inevitably introduce fatigue loading. Now it is vital for the secondary rotors to minimise parasitic drag in the hub region as torque from this will absorb power from the primary rotor that cannot be recovered. It is of no benefit to have a spinner that may deflect the central flow outwards augmenting flow over the inboard blade sections and equally, it is of no benefit to have ducted secondary rotors that produce any flow augmentation.

This is because any augmentation contributes to added thrust (drag) on the spinner or the duct that will consume irrecoverable primary rotor power. This suggests that the secondary rotor system may benefit from having blades of more ideal profile than is usual near the hub centreline not because any very significant gain in secondary rotor power can be obtained but in order to minimise drag in that area. In this scenario the blades would twist to near 90° out-of-plane bringing the blade roots very close each other and to the axis of rotation. The large chord widths nearly parallel to the axis would be

exploited for structural strength of the whole rotor which would most probably use a lot of carbon in its construction and have titanium leading edge erosion protection. Another idea aiming to reduce parasitic drag, perhaps too far-fetched, would be to engineer a rotor generator system with a hollow centre although there would still be issues of drag on the internal surfaces.

**4.6 Secondary rotors on a common axis**

Returning to actuator disc theory, the idea of twin rotors counter-rotating on a common axis enabling a doubling of relative velocity at the generator air gap has been considered, Shen (2007) and Rosenberg (2014). According to simple actuator disc theory, the ideal maximum $C_p$ with the twin rotors in series, assuming they are sufficiently apart for complete pressure recovery near the downstream rotor, increases from the Betz limit of 0 593 only to 0.64, see Newman (1986), or decreases to 0.32 if the swept area of both rotors is accounted. The situation is very different for very lightly loaded secondary rotors

(Fig. 11) where the downstream rotor may operate almost as efficiently as the upstream.

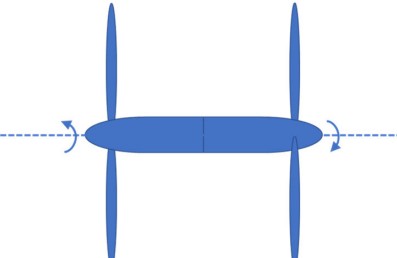

**Figure 11 Twin rotor secondary rotor system**

The potential benefit of a secondary rotor pair in a series arrangement is not only that the design torque and weight of the power train may be reduced compared to a single equivalent rotor but perhaps that a slimmer generator and hence a slimmer

centre-body with less parasitic drag may be realised. Any kind of multi-rotor, secondary rotor system has obvious advantages in torque and weight reduction but having a physical arrangement of support structure and connection to the





primary rotor that minimises parasitic drag will be very important. Based on wind tunnel tests on actuator discs represented as porous screens, Newman (1986) concluded that his theory for multiple actuator discs in series, in the particular case of two actuator discs, became inaccurate only at spacings closer than a disc radius. This gives confidence that at the very low disc loadings applicable to a pair of secondary rotors in series, spaced about a diameter apart, there should be complete

pressure recovery between the rotors. A single rotor of radius 5 m could be replaced by two rotors side by side as in a multi rotor arrangement of radius, $5/\sqrt{2} = 3.536$ m. When the rotors are twins in series on the same axis, the radius required to have the same total thrust at an equivalent axial induction of 0.05 thereby extracting 95% of primary rotor power is related to the velocity recovery approaching the downstream twin. In the analysis following, pressure recovery is assumed and the velocity approaching the downstream turbine is taken as the far wake velocity of the upstream turbine, $\Omega R_0\{1 - \delta\}$ where

the velocity deficit ratio is $\delta$ and would be $2a = 0.1$ for a single ideal actuator disc in inviscid flow. The axial induction factors are selected in an optimisation constrained so that the twin rotors provide the specific total thrust required for primary rotor power extraction and also extract the same total power as a single secondary rotor with the design axial induction value, $a = 0.05$. This is accomplished as follows. The thrust, $t_1$, on a single rotor that would be replaced by the twin system is proportional to the square of the radius, $r_1$, the square of the relative velocity. $V_t = \Omega R_0$ and a thrust coefficient based on

the axial induction $a = a_e = 0.05$. Considering now the equivalent twin rotor system, with axial induction $a_u$ on the upstream turbine, $a_d$ on the downstream turbine, relative velocity $V_t$ on the upstream turbine and $V_t\{1 - 2(1 - z)a_u\}$ on the downstream turbine. The wake velocity deficit ratio is $\delta = 2(1 - z)a_u$ where $z$ is a factor measuring the extent of velocity recovery being 0 when, as for a single actuator disc far wake, the deficit is 2a and 1 if there is complete velocity recovery. For the twin to produce the same total thrust as the single rotor with thrust, $t_1$, requires;

$$r_1^2 a_e(1 - a_e) = r_u^2 a_u(1 - a_u) + r_d^2 a_d(1 - a_d)\{1 - 2(1 - z)a_u\}^2 \tag{20}$$

In addition, if the same total power is required, then, with power being proportional to square of radius, to power coefficient and to cube of relative velocity;

$$r_1^2 a_e(1 - a_e)^2 = r_u^2 a_u(1 - a_u)^2 + r_d^2 a_d(1 - a_d)^2\{1 - 2(1 - z)a_u\}^3 \tag{21}$$

For given values of $z$, Eq. (20) and (21) are solved with the additional assumption that the upstream and downstream rotors have the same radius, $r_u = r_d$, that is to be minimised.




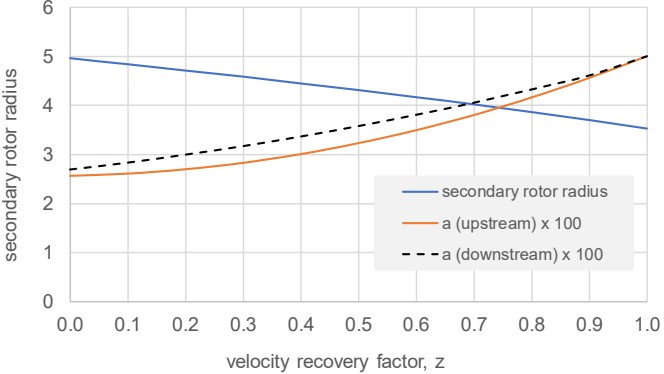

**Figure 12 Secondary rotor radius for no power loss related to velocity recovery factor**

The results in Fig.12 show the variation of secondary rotor radius, upstream rotor induction factor, $a_u$ and downstream rotor induction factor, $a_d$ with velocity recovery factor, $z$. Conventional wake models, such as assessed in a comparative study of

velocity deficit by Luong (2017), suggest little velocity recovery will take place between rotors 2 to 3 radii apart. However such models may be conservative and it is also difficult to gauge their applicability. The very high relative wind speed would imply a very low turbulence intensity which would not assist velocity recovery. However, the loading on the secondary rotors is necessarily very light to avoid too much loss of primary rotor power and the weak wake is may be skewed by centrifugal force. Quite close spacings ~ 1 radius may be beneficial because of the interaction of the rotating wake which is

not accounted in any simple actuator disc modelling. A considerable amount of research into various counter rotating rotor system has taken place since Newman (1986). Tests on a small 6 kW contra-rotating rotor discussed in Shen (2007) indicated that, at the relative high loadings of conventional turbines, 30% more power (as opposed to 8% on the basis of an ideal Cp of 0.593 rising to 0.64) can be obtained. Numerical modelling (also Shen, 2007), of a counter rotating pair of Nordtank 500 kW wind turbines using the EllipSys3D code developed at the Technical University of Denmark (DTU) with

reference to a particular site, predicted 43.5% more energy than for a single turbine. None of the existing literature considers the very light loadings appropriate to a pair of secondary rotors but experiments and CFD analyses generally provide encouragement that performance in real flow will exceed, sometimes greatly exceed, the performance predicted by simple actuator disc inviscid flow models. Even with little velocity recovery where the required diameter of the twin rotors approaches that of a single equivalent rotor, there may be still be net advantage from lighter blade loading, lower generator

torque and reduced generator diameter with associated reduced centre body drag. The velocity recovery that may occur is evidently speculative and may only be better assessed by CFD modelling of a specific design arrangement.





## 5 Concluding remarks

Three quite distinct design directions have emerged from optimisations relating to basic loads predicted by actuator disc theory. These are; a) conventional design with rotor radius predetermined which has been used as a reference, b) the low induction rotor arising from constraint on out-of-plane bending moment and c) the secondary rotor concept arising from
constraint on rotor thrust loading.

In comparison to conventional design, the design challenges in realising a low induction rotor are not radically new. The present work highlights that the power gain in relation to required rotor expansion (a cost) and thrust reduction (a benefit for turbine loads and windfarm wake impacts) is sensitive to the radial distribution of axial induction and discusses optimisation
around these factors. In particular it is shown that the same power gain of 7.6% with an optimum radially constant axial induction of 0.2 that required a rotor expansion of 11.6% can be achieved with an expansion of only 6.7% when the axial induction varies radially and is optimised. The modelling developed here enables definition of a space of all self-consistent combinations of power gain, rotor expansion and thrust reduction with each associated axial induction distribution. This could enable a preliminary determination of an overall optimum axial induction distribution using a combined wind turbine
and wind farm cost of energy model. Structural design issues of a low induction rotor have been reviewed by Chaviaropoulos, (2014). An expanded rotor of standard design could be operated at low induction using pitch control thereby restricting the steady state blade root bending moment but this would not be satisfactory. It is vital to contain all loads of the expanded rotor, steady state, dynamic and loads when idling in extreme wind conditions by limiting the lift and drag of the rotor to the levels of the non-expanded reference rotor. This calls for lower lift aerofoils or reduced solidity or
both. There is much less of a design challenge in the low induction rotor with a radially varying optimised axial induction distribution (Fig. 4.) as compared to the constant induction of 0.2. The required rotor expansion is much less and the progressive reduction of axial induction towards the blade tip is sympathetic to blade structural design with a natural taper in strength and solidity from rotor to tip. The graded reduction in spanwise axial induction is also much more favourable than a global reduction to 0.2 for limiting tip deflection to maintain acceptable tower clearance without having undue added cost in
stiffening the blade.

Secondary rotors have not been used on an operational wind turbine although a design is now being developed (Leithead, 2018). As being very new territory for wind turbine design, secondary rotor design is discussed somewhat more extensively. The main aim in using secondary rotors is to have a drive train with much reduced design torque compared to the usual
transmission system based on power take-off from a central shaft. That can certainly be achieved with torque reduction of 1 to 2 orders of magnitude being possible depending on specific design choices. Although the design of secondary rotors is much more demanding than of conventional rotors of the same diameter, the design torque reduction is so great that it seems certain that substantial savings in drive train cost can be realised. The focus of the secondary rotor design exploration is on



VAWTs as the primary rotor rather than HAWTs because it solves a key problem with VAWTs of relatively low shaft speed leading to high drive train torque and expensive drive trains whereas, as applied to HAWT design, it could introduce major problems for primary rotor blade design. It emerges that the radial distribution of axial induction is not directly important for secondary rotor design as all distributions with the same area averaged axial induction will lead to the same size of

secondary rotor. The high relative wind speed compensates for relatively small rotor diameter and very low design axial induction in a way that for primary rotors in the multi-megawatt range maintain a Reynolds number $\sim 10^6$ and suggests a solidity only a little higher than that typical of large HAWTs is required. However with secondary rotors, very high tip speeds are desirable to limit drive train torque and to limit the overall scale of rotor and generator system. Also steady state operational loads are exceptionally high in relation to rotor diameter. Having multiple secondary rotors (more than 1 per

blade) has the usual benefits of multi rotors (Jamieson 2011) in reducing net torque, weight and cost of secondary rotor systems but, as was mentioned, it is particularly important with secondary rotors to minimise losses from parasitic drag or degradation of primary blade performance depending on their physical mounting arrangement. The idea of realising multi-rotors as a twin set in series on a common axis looks promising and may have mileage considering the very low axial induction levels required of secondary rotors to avoid wasting primary rotor power. Whether this is a particularly good idea

cannot be resolved without evaluating specific design arrangements and developing greater understanding of the flow field around the secondary rotor system as a twin pair.

The preliminary evaluation of the X-rotor VAWT design (Leithead 2019) suggests that use of secondary rotors will lead to more competitive VAWT designs. Another innovative VAWT design, the DeepWind VAWT of Paulsen (2015) has major

savings through integration of rotor blades shaft and support structure into a single element. On the other hand, substantial challenges remain for the design and maintenance of the underwater electrical generating system. Could an adapted variation of this design with modular secondary rotors that can form a more economical power train to be accessed and maintained above sea level be advantageous?

In summary, three quite different rotor optimisations are shown to arise naturally from long established actuator disc equations and can usefully guide high level design of the innovative rotor systems described as "the low induction rotor" and the "secondary rotor".

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
