# Peer review of "Top Level Rotor Optimisations based on Actuator Disc Theory"

_Wind Energy Science, 2019_

## Referee Comment (RC1) · Anonymous Referee #1 · 10 Dec 2019

An interesting paper addressing different rotor design ideas and how the low induction rotor may help to further reduce the cost of energy when used as tip rotors. The paper has a nice introduction to classical wind turbine rotor aerodynamics and showing that the power production can be made higher by allowing the radius to be increased, but keeping the same root bending moment. Doing this the axial induction factor is decreased from 0.33 to a=0.2. This result is known but is taken further by searching for a more optimum distribution for the axial induction factor a(x). This is done by a formal optimization and describing a(x) with two parameters that together describe smooth and realistic distributions. The last part of the paper is very interesting since it describes and discusses various innovative ideas how to use these low induction rotors, especially on VAWTs. It would be nice if the derivation of equation (10) on the

bottom of page 10 is given.

---

## Referee Comment (RC2) · Anonymous Referee #2 · 23 Mar 2020

This paper makes a nice addition to the body of work on novel designs to reduce cost of energy, through low induction rotors and secondary rotor design. The design tradeoffs for these concepts are well explained with classical momentum theory.

My main feedback is this paper needs to be better organized, which can primarily be achieved with a more robust introduction. Half of the paper is dedicated to VAWTs and secondary rotor design, but this is not mentioned in the introduction. The final paragraph in Section 2, discussing Figure 2, which provides the motivation for secondary rotors, is very out of place. This content could be moved to the introduction and the beginning of Section 4 as a transition from Section 2/3. Discussion of how the results and analysis fit into the existing body of literature is scattered throughout the paper, but a more traditional discussion of the existing literature in the introduction would likely be

more effective.

---

## Author Comment (AC2) · 27 Mar 2020

Thank you for your comments. I will respond regarding derivation of Equation (10)with a few lines of introduction which should clarify the derivation "The power produced by the primary (VAWT) rotor is $P=0.5U(0)ËĘ3$ x $2R(0)$ x $L$ x $C(P)$ and the total power extracted by n secondary rotors is $p=np(n)=P(1-a)$. For each secondary rotor, $p(n)=0.5(âĎę xR(0))ËĘ3$ x pi x $r(n)ËĘ2$ x $4a(1-a)$. Hence the ratio of radius of one of n secondary rotors to that of the primary rotor can be expressed as; " .............. Eq 10 follows. Here I have bracketed some characters which will appear as subscripts in the text proper.

---

## Author Response (AR1)

**Author's Response**

In response to referee 1, I have added a few lines at the start of Section 4.2 which I hope makes clear the derivation of Equation 10.

In response to referee 2, I have removed out of place text using some of it in a new introduction section which also has much additional new text in order to set the full background. In support of this a new reference (to Snel) is added.

[revised manuscript text omitted]

---

## Author Response (AR3)

Thank you for your comments. I will respond regarding derivation of Equation (10)with a few lines of introduction which should clarify the derivation "The power produced by the primary (VAWT) rotor is P=0.5U(0)ËȨ3 x 2R(0) x L x C(P) and the total power extracted by n secondary rotors is p=np(n)=P(1-a).  For each secondary rotor, p(n)=0.5(âĎę xR(0))ËȨ3 x pi x r(n)ËȨ2 x 4a(1-a).  Hence the ratio of radius of one of n secondary rotors to that of the primary rotor can be expressed as; " ............. Eq 10 follows. Here I have bracketed some characters which will appear as subscripts in the text proper.

[Figure]

[Figure]

I fully see the force of your comments having read through afresh and have I hope addressed all your points. I have made major revisions creating an introduction section which outlines the structure of the paper basically as in three sections a) demonstrating optimizations from basic AD theory leading to low induction and secondary rotors b) more extended discussion of low induction and c) more extensive discussion of secondary rotors based VAWT primary as most promising application. Within the introduction I have outlined the background in each area and included most of the references. Only the references to the twin rotor concept which emerges later as an option for secondary

rotors are left in place as it is most natural to have them there and it is not a central theme of the paper.

[revised manuscript text omitted]

**Response to Associate Editor's Review**

Thanks for your comments which are spot on and very helpful.

Although revisions were "minor" in the sense that nothing has changed in the essential analytical work of the paper, I felt the best response was to have another go at the introduction section. New additions and edits are not that many but text has been moved around and replaced in a way that I hope better meets your criticisms.

I have deleted the bullets points that were out of place at start of introduction and replaced with a sentence at the end. I have included the references you suggested and feel stupid to have missed them!. Pity that Chris had not published a paper on his optimisations as after your reminder I recall discussions with him at the WESC of 2017.

**Top Level Rotor Optimisations based on Actuator Disc Theory**

Peter Jamieson[1]

[1]Centre for Doctoral Training in Wind and Marine Energy, University of Strathclyde, Glasgow, G1 1XW, UK

*Correspondence to*: P Jamieson (peter.jamieson@strath.ac.uk)

5   **Abstract**. Ahead of the elaborate rotor optimisation modelling that would support detailed design, it is shown that significant insight and new design directions can be indicated with simple, high level analyses based on actuator disc theory. The basic equations derived from actuator disc theory for rotor power, axial thrust and out of plane bending moment in any given wind condition involve essentially only the rotor radius, $R$, and the axial induction factor, $a$. Radius, bending moment or thrust may be constrained or fixed with quite different rotor optimisations resulting in each case. The case of fixed radius or rotor

10   diameter leads to conventional rotor design and the long-established result that power is maximised with an axial induction factor, $a = 1/3$. When the out of plane bending moment is constrained to a fixed value with axial induction variable in value (but constant radially) and when rotor radius is also variable, an optimum axial induction of $1/5$ is determined. This leads to a rotor that is expanded in diameter 11.6% gaining 7.6% in power and with thrust reduced by 10%. This is the "low induction rotor" which has been investigated by Chaviaropoulos (2013). However, with an optimum radially varying

15   distribution of axial induction, the same 7.6% power gain can be obtained with only 6.7% expansion in rotor diameter. When without constraint on bending moment, the thrust is constrained to a fixed value, the power is maximised as $a \to 0$ which for finite power extraction would require $R \to \infty$. This result is relevant when secondary rotors are used for power extraction from a primary rotor. To avoid too much loss of the source power available from the primary rotor, the secondary rotors must operate at very low induction factors whilst avoiding too high a tip speed or an excessive rotor diameter. Some general

20   design issues of secondary rotors are explored. It is suggested that they may have most practical potential for large vertical axis turbines avoiding the severe penalties on drive train cost and weight implicit in the usual method of power extraction from a central shaft.

**1 Introduction**

**1.1 General background**

25   Two quite different innovative rotor concepts have been considered previously. These are the low induction rotor and the secondary rotor.

A low induction rotor in optimal operation is designed to operate with lower values of axial induction than 1/3, the ideal value according to the according to basic actuator disc theory to maximise power at a fixed chosen diameter. The primary motivation for the low induction concept is to lower the cost of energy in scenarios where sacrificing some power in

reducing design induction values below 1/3 leads to relatively more significant load reductions that are of overall economic benefit to the design. It has long been recognised in commercial designs and in windfarm operation that comparatively small reductions in design or operating induction from 1/3 can be beneficial. This is incorporated in public designs such as the DTU 10 MW reference turbine further developed as an IEA reference turbine by Borlotti (2019) while the potential benefits of yet more radical reductions in induction to around 0.2 had been highlighted by Chaviaropoulos (2013).

The secondary rotor concept involves extracting power using a rotor generator system mounted on the blades of an otherwise conventional primary turbine. The secondary rotors operate at high speed in much elevated relative air speeds leading to much smaller and lighter power conversion equipment than with a conventional centre shaft based drive train. This idea is much older than the low induction idea and arose in designs such as the airborne system of Jack () and spaceframe turbine Watson () 
[revised manuscript text omitted]